# Therapeutic Vaccines for Non-Communicable Diseases: Global Progress and China’s Deployment Pathways

**DOI:** 10.3390/vaccines13080881

**Published:** 2025-08-20

**Authors:** Yifan Huang, Xiaohang Lyu, Yiu-Wing Kam

**Affiliations:** Division of Natural and Applied Science, Duke Kunshan University, No. 8 Duke Avenue, Kunshan 215316, China; yifan.huang@dukekunshan.edu.cn (Y.H.); xiaohang.lyu@dukekunshan.edu.cn (X.L.)

**Keywords:** non-communicable diseases (NCDs), therapeutic vaccines, vaccine development, immunotherapy, public health policy, health equity, implementation challenges

## Abstract

Background: Non-communicable diseases (NCDs) have become a major threat to global public health, with the disease burden particularly severe in developing countries, China being one of them. The preventive and control effects of traditional treatment methods on NCDs are limited, and innovative strategies are urgently needed. In recent years, vaccine technology has expanded from the field of infectious diseases to non-communicable diseases (NCDs). Therapeutic vaccines have shown the potential to intervene in chronic diseases through immunomodulation, but their research and development (R & D), as well as promotion, still face multiple challenges. Methods: This article systematically reviews the current development status of NCD vaccines worldwide and points out the imbalance in their matching with disease burden: current research focuses on the field of cancer, while there is a lack of targeted vaccines for high-burden diseases such as hypertension and chronic kidney disease; the progress of independent R & D in China lags behind, and there are implementation obstacles such as uneven distribution of medical resources between urban and rural areas and low public willingness to be vaccinated. Results: By analyzing the biological mechanisms of NCD vaccines and non-biological challenges, phased solutions are proposed: In the short term, focus on target discovery and improvement of vaccine accessibility. In the medium term, strengthen multi-center clinical trials and international technology sharing. In the long term, build a digital health monitoring system and a public–private partnership financing model. Conclusions: The breakthrough of NCD vaccines requires interdisciplinary collaboration and systematic policy support. Their successful application will reshape the paradigm of chronic disease prevention and control, providing a new path for global health equity.

## 1. Introduction

Non-communicable diseases (NCDs) refer to a group of chronic health conditions that, unlike communicable diseases, cannot be transmitted directly from person to person. Unlike infectious diseases caused by pathogens such as bacteria or viruses, NCDs are often associated with shared risk factors, including tobacco use, physical inactivity, excessive alcohol consumption, unhealthy diets, and air pollution [1]. According to the Global Burden of Disease (GBD) 2021 data, NCDs pose a major global health challenge, accounting for more than half of the worldwide disease burden, underscoring their critical impact on public health. This burden is particularly pronounced in China, where NCDs are responsible for 90.99% of total deaths and 86.74% of total disability-adjusted life years (DALYs), highlighting their significant role in the country’s overall disease burden. Specifically, stroke and ischemic heart disease contributed 13.23% and 8.87% of DALYs in China, compared to 5.57% and 6.55% globally. Moreover, mortality data from China revealed that stroke and ischemic heart disease accounted for 22.16% and 16.73% of total deaths, significantly surpassing the global proportions of 13.25% and 10.68%, respectively [2] (Table 1). In addition to cardiovascular diseases, chronic obstructive pulmonary disease (COPD) is also a major contributor to disease burden and mortality in China, which ranks third in both DALYs and deaths, with 5.88% of DALYs and 10.99% of deaths, compared to 2.77% and 5.48% globally. Other NCDs such as Alzheimer’s disease, diabetes mellitus, and stomach cancer feature prominently among the top ten contributors to both DALYs and deaths in China and around the world [2] (Table 1). Looking ahead, projections from the GBD further suggest that, from 2022 to 2050, the prevalence of poor health and early death caused by infectious diseases will gradually decline, while the burden of NCDs is expected to rise. This trend emphasizes the growing significance of NCDs as a global health priority [3]. Consequently, NCDs are emerging as one of the most pressing public health challenges of the 21st century.

Despite the substantial global and domestic burden of NCDs, current treatments remain largely focused on traditional approaches such as lifestyle modifications, medication, and, in some cases, surgical interventions [4,5]. However, the ongoing rise in NCD prevalence, coupled with concerns about treatment efficacy and the overall well-being of patients, highlights the urgent need for innovative strategies to more effectively prevent and manage these diseases [6,7,8].

Traditionally used to combat communicable diseases, vaccination is now being considered as a potential strategy for addressing NCDs [9]. While vaccines have long been a cornerstone of infectious disease management, they have also been recognized as a tool that not only promotes general health and well-being but also enhances economic productivity. Based on this insight, the concept of NCD vaccines seems necessary to be brought to the table. Unlike conventional vaccines that target pathogens or infected cells, NCD vaccines are designed to target specific cells, proteins, or other molecules involved in the development of non-communicable diseases. These vaccines, which include both prophylactic and therapeutic types, present an opportunity to prevent disease more effectively compared to traditional NCD treatments [10]. Furthermore, NCD vaccines offer the advantage of providing long-lasting immune protection, reducing patients’ reliance on ongoing medications or disease management strategies and thereby improving overall quality of life. Although the initial development and promotion of NCD vaccines can be costly [11], their ability to prevent diseases at an early stage and their “once-and-for-all” therapeutic potential could significantly lower the long-term costs associated with chronic medication use and complex surgical interventions [12].

This review examines the development and implementation of NCD vaccines globally, with a special focus on China. Given that the research on and clinical application of NCD vaccines are still in their early stages worldwide, the review critically assesses whether current NCD vaccines align with China’s unique disease burden. Additionally, it explores the key challenges associated with the deployment of NCD vaccines in China and provides targeted recommendations. These include a proposed timeline for future initiatives, which may also serve as a reference for other developing nations.

## 2. Methodology

This study adopted a narrative and descriptive literature review design to examine the global landscape of NCD vaccine development, with a particular focus on China’s research activity, market dynamics, and policy orientation. Cases from representative low- and middle-income countries (LMICs) were also included. The goal was to synthesize available evidence from academic publications, regulatory databases, clinical trial registries, and market intelligence reports to identify trends, challenges, and gaps in NCD vaccine development and deployment.

A structured search was conducted using combinations of keywords such as “NCD vaccines,” “therapeutic vaccines,” “cancer vaccines,” “cardiovascular vaccines,” “chronic diseases,” “China,” and “therapeutic vaccines clinical trials” across academic databases including PubMed, Scopus, and Google Scholar, with an emphasis on peer-reviewed studies published between 2015 and 2025. Only English-language publications were included. For the search on existing therapeutic NCD vaccines, only information available up to 8 July 2025 was included to ensure consistency and accuracy in data comparison. In addition to academic publications, official government and regulatory databases were reviewed to provide a comprehensive and up-to-date understanding of the policy and development environment. These included ClinicalTrials.gov, the Chinese Clinical Trial Registry (ChiCTR), the National Medical Products Administration (NMPA) website, and even pharmaceutical companies’ official websites, which offered insights into registered trials, approvals, and domestic R & D initiatives. National policy documents such as the Healthy China 2030 initiative and the Medium- and Long-Term Plan for Chronic Disease Prevention and Treatment (2017–2025) were also examined to assess strategic priorities. Furthermore, to explore the funding dynamics and market prioritization of NCD vaccines in contrast to other NCD interventions, market intelligence reports from recognized commercial research firms, such as Grand View Research, and McKinsey, were consulted, provided that their methodology and data sources were clearly documented.

All selected sources were internally reviewed by research team members to ensure their relevance, accuracy, and credibility, and only those published by official agencies, reputable journals, or widely recognized research organizations were included.

## 3. Existing NCD Vaccines

NCD vaccines can be classified into two types based on their objectives: prophylactic vaccines and therapeutic vaccines. Current studies and available data predominantly focus on therapeutic NCD vaccines. The complexity of NCDs arises from a range of contributing factors, including genetic predisposition, immune system function, exposure to infectious agents, and lifestyle choices, among others. This intricate etiology presents a significant challenge to identifying specific targets for prophylactic vaccine development, creating obstacles in designing vaccines with precise and effective mechanisms of action.

Therapeutic vaccines, as their name implies, are administered after a disease or infection has developed. Their primary goal is to activate an immune response tailored to recognize and eliminate infected or abnormal cells [13]. Cancer vaccines represent the most researched area within therapeutic NCD vaccines, although other conditions like hypertension and Alzheimer’s disease are also being investigated [14,15]. Most of these vaccines remain in the clinical research stage (Table 2). Among currently developed therapeutic vaccines, the lung cancer vaccine is the most advanced. One such example is CimaVax-EGF, which was developed in Cuba and has undergone comprehensive clinical trials in selected countries. It is a therapeutic vaccine for non-small cell lung cancer (NSCLC) that works by inhibiting the action of epidermal growth factor (EGF), a key driver of cancer cell proliferation [16]. Specifically, the vaccine stimulates a B-cell response to produce antibodies that bind to and neutralize EGF in the body, preventing it from interacting with receptors on the surface of cancer cells [17]. As a result, the proliferation of cancer cells is slowed, effectively managing disease progression. CimaVax-EGF has received approval for medical use in some regions [17].

Most therapeutic NCD vaccines share a similar underlying mechanism of action, aiming to combat disease by activating a specific immune response. They work by inducing or enhancing cell-mediated immunity, typically through the stimulation of cytotoxic T-cells or the activation of B-cells to produce specific antibodies. The ultimate goal is to remodel the host’s immune system to eliminate the disease and establish long-term immunological memory [15]. Provenge, the first therapeutic vaccine approved by the FDA in 2010, is a prominent example [41]. It utilizes a patient’s own antigen-presenting cells (APCs) to present prostate cancer-associated antigens (PAPs) to T-cells, thereby activating the immune system to target and attack prostate cancer cells [42].

By comparing and analyzing Table 1 and Table 2, it becomes clear that therapeutic NCD vaccines currently under development, research, or clinical trial largely align with the top 10 diseases contributing to the global and Chinese NCD burden. To date, therapeutic NCD vaccines that have been approved for clinical use include CimaVax-EGF (for non-small cell lung cancer), Sipuleucel-T (for prostate cancer), and Inclisiran (for dsylipidemia). Other therapeutic cancer vaccines, such as OncoVAX for stage 2 colon cancer, are still in the clinical trial phase but show considerable promise in reducing the risk of cancer recurrence and improving patient outcomes [43]. A renal cell carcinoma vaccine, NeoVax, has also been proven to be safe and efficacious for postoperative adjuvant therapy in its preliminary clinical trials [24].

While therapeutic cancer vaccines have seen relatively advanced development, chronic conditions related to these cancers remain largely under-addressed. For instance, chronic obstructive pulmonary disease (COPD) is a significant independent risk factor for lung cancer, doubling a patient’s likelihood of developing the disease [44,45]. However, there are currently no vaccines specifically targeting COPD, leaving a critical gap in therapeutic strategies and placing a heavy burden on global healthcare systems. A similar issue exists with other chronic diseases linked to cancer, such as chronic liver disease and chronic kidney disease. It is widely recognized that liver cancer often develops from a cirrhotic or heavily fibrotic liver [46]. While therapeutic vaccines for kidney cancer, such as GPC3, have entered clinical trials, vaccines specifically targeting chronic kidney disease remain poorly researched. Chronic kidney disease (CKD), a significant contributor to NCD-related deaths, can lead to kidney cancer via mechanisms such as oxidative stress caused by a uremic environment or pre-existing cystic conditions [47]. Currently, vaccines available for CKD only offer indirect prevention (e.g., influenza and pneumococcal vaccines), rather than direct therapeutic solutions, highlighting a significant unmet need in this area. Although in recent years there have been some potential vaccine treatments for COPD and CKD, they remain in the very early stages of research, with insufficient data and trials to support their credibility, still leaving the therapeutic gap largely unaddressed [48,49]. Furthermore, other chronic non-communicable diseases that are not traditionally seen as NCDs, such as hearing loss, low back pain, and depression, remain without therapeutic vaccines, even though they account for a substantial portion of global disability-adjusted life years (DALYs) attributed to NCDs.

Nevertheless, there is also good news. The development of vaccines for non-cancer chronic diseases has been gaining momentum in recent years. For example, a vaccine for dyslipidemia, a condition closely linked to increased cardiovascular risk, is now available. Inclisiran is currently the only siRNA-based cholesterol-lowering drug approved for market use globally [24]. It treats dyslipidemia by using RNA interference (RNAi) to silence specific genes through the degradation of target mRNA, thereby preventing the production of proteins that contribute to elevated cholesterol levels [50,51]. Similarly, several vaccines for diabetes are also progressing in R & D [52]. CYT013-IL1bQb, a vaccine targeting type 2 diabetes mellitus, completed its phase I/IIa study in 2011 in Germany and Switzerland [37]. It uses virus-like particles (VLPs) to deliver a low-activity mutant of IL-1β, breaking immune tolerance and inducing sustained neutralizing antibodies to inhibit IL-1β-driven inflammatory pathways. This mechanism offers a potential long-lasting therapeutic approach for type 2 diabetes and other IL-1β-dependent diseases such as gout and autoinflammatory syndromes [28]. Another vaccine targeting type 1 diabetes mellitus, BHT-3021, which completed its phase I trial in the United States, Australia, and New Zealand [53], gained its effect through delivering a proinsulin-encoding DNA plasmid to antigen-presenting cells, leading to antigen presentation without co-stimulation and subsequent induction of immune tolerance in proinsulin-reactive T-cells [54].

Examining global therapeutic NCD vaccine distribution reveals a significant concern: uneven development between China and other parts of the world. Despite the high prevalence of NCDs in China, there is insufficient research, development, and introduction of therapeutic vaccines to address these diseases, which could severely impact the health and well-being of Chinese society. Among the therapeutic vaccines for major NCDs listed in Table 2, none of them have been self-developed in China. A large proportion of these listed therapeutic NCD vaccines are developed by Western developed countries that are already privileged, rather than developing countries or LMICs. What is more, when most of these therapeutic vaccines have already undergone clinical trials or even reached the market in other countries, there remains no record in China of their introduction or registration for clinical trials, manifesting a clear delay in China’s participation in the global advancement of therapeutic NCD vaccines.

Additionally, national priorities in NCD vaccine research differ, leading to asymmetric development across disease areas in different countries. For instance, the United States leads in cancer vaccine research, while Switzerland has made significant advancements in cardiovascular disease and other NCD vaccines (Table 2). These disparities may result from differences in technological capabilities, financial investments, institutional support, societal attitudes, and healthcare priorities. To foster more balanced global progress, it is essential to analyze and align deficiencies in science, policy, finance, and public awareness to support mutual learning and collaboration, especially for countries like China and other developing nations.

## 4. Challenges to NCD Vaccine Development and Implementation

NCD vaccines have the potential to offer patients long-lasting immune protection, reducing the need for ongoing medication and thereby decreasing the cumulative costs associated with long-term treatments. However, the path to developing and implementing these vaccines is riddled with significant challenges at both the development and deployment stages (Figure 1).

## 5. Developmental Challenges

The development of vaccines for NCDs presents numerous challenges, stemming from the complexity of these diseases as well as issues related to funding, safety, and cost-effectiveness. One of the primary obstacles is the multifactorial nature of NCDs. Unlike infectious diseases, which are caused by specific pathogens, NCDs are driven by a combination of genetic, environmental, and lifestyle factors, making it difficult to pinpoint a single target for intervention [55]. For example, cardiovascular disease—the leading cause of death globally—is associated with multiple risk factors, including unhealthy diets, physical inactivity, and smoking [56]. Similarly, type 2 diabetes is heavily linked to lifestyle factors, while Alzheimer’s disease is influenced by a complex interplay of genetic predisposition, lifestyle, and environmental factors [57,58]. This multifaceted etiology makes prevention and treatment far more challenging than for infectious diseases. As a result, many therapeutic vaccine trials for NCDs struggle to demonstrate sufficient efficacy and are often terminated or discontinued (Table 2). For instance, AngQb-Cyt006, a vaccine designed to treat hypertension, failed to progress to phase III trial due to its lower antihypertensive effects compared to existing RAS inhibitors [59]. Another example is therapeutic vaccines for Alzheimer’s disease. Clinical trials for CAD 106 were terminated because of negative results observed in other anti-amyloid therapies for Alzheimer’s disease, leading sponsors to believe that continued development might not be worthwhile [60].

Regulatory and safety concerns further compound the challenges of NCD vaccine development. Unlike infectious disease vaccines, which can demonstrate protective effects within weeks or months, NCD vaccines target conditions such as cardiovascular disease, cancer, and metabolic disorders that develop gradually over many years. In a real-world study, a single dose of an mRNA COVID-19 vaccine showed its effectiveness against SARS-CoV-2 infection within just 7 weeks, demonstrating that infectious disease vaccine efficacy can be observed within weeks to months after vaccination [61]. NCD vaccines, in contrast, require long-term follow-up to demonstrate clinical benefit. In a phase III trial of BiovaxID, a personalized idiotype vaccine for follicular lymphoma, nearly five years of follow-up passed before efficacy emerged [62]. Moreover, validated surrogate endpoints for many NCDs remain limited, further prolonging evaluation timelines [63]. This extended period for demonstrating efficacy, combined with the rigorous safety requirements for chronic-use interventions, strains resources and complicates regulatory approval, resulting in a slower, more complex, and costlier path to deployment.

These challenges are further intensified by the difficulty of establishing cost-effectiveness. Vaccines for infectious diseases, such as measles and polio, have demonstrated high effectiveness and efficiency in reducing disease burden, particularly in low-income settings, with substantial returns on investment from widespread immunization campaigns [63,64]. In contrast, vaccines for NCDs often struggle to demonstrate similar levels of cost-effectiveness. Non-vaccine interventions, including adopting healthier diets, increasing physical activity, and quitting smoking, have shown a far more significant impact on reducing the risk of certain NCDs, such as cardiovascular diseases. These measures are not only more effective in many cases but also typically more affordable. This raises critical questions about whether investing in NCD vaccines is the most efficient use of limited health resources, particularly when alternative strategies may offer greater impact at lower cost [65,66].

The extended clinical timelines required to demonstrate preventive impact, coupled with comparatively low cost-effectiveness versus infectious disease vaccines or behavioral interventions, result in NCD vaccines receiving relatively low funding and policy prioritization. Global health agendas, especially during emergencies such as the COVID-19 pandemic, tend to channel substantial resources toward infectious disease control. For example, the U.S. government invested USD 31.9 billion into mRNA COVID-19 vaccine development, leveraging decades of foundational research [67,68]. In contrast, NCD vaccines receive far less attention and funding, with only 1–2% of development assistance allocated to their financing, and have stagnated for the last three decades [69]. This disparity makes it difficult to advance NCD vaccine research, which involves high costs for preclinical studies, clinical trials, and regulatory approvals. The termination of the phase II clinical trial of HER-Vaxx is one such example, which was due to a lack of organizational prioritization [32]. Furthermore, the long timeline for measurable outcomes of NCD vaccines complicates justifying large-scale investments, particularly in a funding landscape that prioritizes immediate results.

This global trend is also prominent in China. While China has prioritized NCD prevention and control in major policy frameworks such as Healthy China 2030 and the Medium- and Long-Term Plan for Chronic Disease Prevention and Treatment (2017–2025), setting ambitious targets for reducing premature mortality and improving health literacy, none of them mentioned vaccines as a strategy for NCD risk reduction (e.g., cancer, diabetes, cardiovascular disease) [70,71]. In cancer vaccines specifically, a comparative review of clinical trials registered between 2014 and 2024 found only 89 trials in China versus 757 in the United States—just 2.2 % of the global total—and Chinese trials covered only about five cancer types, compared with more than twenty in U.S. trials [58]. According to Grand View Research, the overall Chinese cancer immunotherapy market (including monoclonal antibodies, checkpoint inhibitors, CART, and cancer vaccines) generated around USD 4.6 billion in 2023, with cancer vaccines accounting for only USD 560.7 million (~12% of the total) [72,73]. Together, the limited scope of clinical trials and the small share of investment clearly underscore China’s lack of interest in NCD vaccines.

In summary, the challenges of developing vaccines for NCDs highlight the complexity of addressing chronic diseases through traditional vaccine approaches. Overcoming these obstacles will require a more holistic public health strategy that balances vaccine development with other effective prevention methods, alongside increased and more equitable funding for NCD-related research and innovation.

## 6. Implementation Challenges

Even after NCD vaccines are developed, their successful implementation faces a series of equally formidable challenges. While overcoming scientific and regulatory hurdles is critical, the true measure of success lies in effectively integrating these innovations into health systems and ensuring their uptake within the communities that stand to benefit most. The transition from R&D to practical application presents unique difficulties, such as determining the ideal timing for vaccination and addressing societal and infrastructural barriers. Successfully navigating this complex implementation process requires a comprehensive, multifaceted strategy that considers factors ranging from public perception to equitable healthcare access.

One of the primary challenges lies in identifying the optimal timing for vaccination. Unlike infectious diseases, which often present acute and evident symptoms, NCDs such as cardiovascular diseases, cancers, and diabetes develop slowly over many years, often without noticeable symptoms until they reach advanced stages. For instance, many cancers, such as colon, prostate, or breast cancer, typically remain undetected until they progress to later stages. This gradual and often subtle onset complicates early detection and preventive interventions, making it difficult to determine when vaccination would be most effective. In regions such as China, the lack of robust early screening infrastructure further exacerbates the problem. Many health systems lack routine monitoring mechanisms needed to identify key risk indicators for these conditions, creating significant obstacles to targeting appropriate populations for vaccination and successfully deploying NCD vaccines [74]. A similar situation is seen in Nigeria, where screening assessment for breast, cervical, and colon cancers—the major contributors to cancer morbidity in the region—remains extremely low [75], highlighting how insufficient screening capacity in many LMICs hinders early detection and the feasibility of vaccine-based interventions.

Another major implementation barrier is vaccine hesitancy, particularly for diseases that are not perceived as immediate health threats. Vaccines aimed at preventing NCDs like cancer or cardiovascular diseases may not carry the same sense of urgency as those for acute infectious diseases like influenza or measles. In part, this is due to prevailing societal perceptions that NCDs are long-term conditions associated with aging or lifestyle choices, reducing the perceived necessity for vaccination. Concern about vaccine safety and potential side effects also fuels hesitancy, even in the presence of evidence demonstrating they are well tolerated and effective [76,77]. In China, public confidence in vaccines has been further undermined by high-profile incidents of adverse side effects reported over the past decade [78]. Additionally, experiences during the COVID-19 vaccination rollout in developing countries demonstrate how social norms and healthcare workers’ attitudes influence vaccine acceptance. In India, low awareness and hesitancy among healthcare professionals (HCPs), along with insufficient vaccine recommendations from HCPs, have been identified as major barriers to administering COVID-19 vaccines to adults with NCDs [79].

Issues related to access and equity present another substantial barrier to the successful implementation of NCD vaccination programs. In countries like China, the urban–rural divide in healthcare infrastructure remains a persistent obstacle, with rural and underdeveloped areas often lacking basic services, including vaccination programs [80]. This disparity is compounded by the high cost of NCD vaccines, which may be unaffordable for low-income or rural populations. Even with increased government investment aimed at reducing the financial burden of these vaccines, resources are often insufficient or unevenly distributed, leaving marginalized groups at greater risk of being excluded from preventive healthcare [81]. This deepens existing health inequities and exacerbates disparities in NCDs outcomes, preventing broader public health goals from being achieved.

Implementation challenges highlight the multifaceted difficulties of translating scientific advancements into large-scale public health applications. While the development of NCD vaccines is a critical step forward, their successful implementation will require addressing not only scientific and regulatory obstacles but also societal, infrastructural, and policy-related barriers. Overcoming these barriers will demand coordinated action from governments, healthcare systems, and communities alike to ensure that the promises of these vaccines are realized equitably and effectively.

## 7. Recommendations

To address the multifaceted challenges in the research, development, and implementation of NCD vaccines outlined above, this paper proposes a comprehensive set of solutions aimed at advancing the long-term development of the NCD vaccine field and driving progress in global public health (Figure 2).

### 7.1. Biological

One of the main challenges in developing vaccines for non-communicable diseases (NCDs) lies in identifying precise molecular targets capable of effectively modulating immune responses. Unlike traditional vaccines for infectious diseases, which aim to stimulate immunity against external pathogens, NCD vaccines must focus on addressing dysregulated immune pathways that drive chronic disease progression. A key strategy in this effort is the identification of disease-specific biomarkers, including proteins, enzymes, and immune checkpoints, that can serve as therapeutic targets. Immune checkpoints, such as PD-1/PD-L1 and PD-L2, play a critical role in maintaining immune tolerance and preventing excessive immune activation. Emerging research suggests that modulating these pathways could enhance immune responses against tumor cells in cancer, and this approach shows promise for NCDs like atherosclerosis, neurodegenerative disorders, and metabolic diseases [82]. For example, targeting the PD-1/PD-L1 axis has demonstrated potential in cardiovascular diseases by reducing chronic inflammation and helping to prevent plaque formation in atherosclerosis [83]. Similarly, in Alzheimer’s disease, immune checkpoint pathways are involved in regulating neuroinflammation, where excessive immune suppression may exacerbate disease progression [84]. Integrating immune checkpoint modulation into vaccine R & D could facilitate the induction of protective immune responses to counteract the pathological immune dysfunction seen in NCDs.

In addition to immune checkpoint modulation, artificial intelligence (AI) is emerging as a powerful tool to overcome the challenges of target discovery in NCD vaccine development. AI-driven platforms, such as in silico Medicine’s Evo 2, leverage deep learning and multi-omics integration—bringing together genomics, transcriptomics, proteomics, and single-cell data—to identify novel disease-associated biomarkers. This systems biology approach enables the detection of subtle molecular patterns and dysregulated pathways, offering new opportunities to identify viable vaccine targets for complex conditions such as neurodegeneration, cardiovascular diseases, and metabolic disorders [85]. By uncovering hidden molecular drivers of chronic inflammation and tissue dysfunction, platforms like Evo 2 can accelerate the rational design of immunotherapies and therapeutic vaccines, advancing the fight against NCDs.

Apart from the discovery of novel antigens, repurposing existing antibody therapeutics offers a promising avenue for developing vaccines targeting non-communicable diseases (NCDs). Monoclonal antibodies, widely used in NCD drug therapies, provide crucial insights into immune targets that can be harnessed for active immunization strategies [86]. By designing vaccines that stimulate the body’s own production of antibodies against disease-associated proteins, such as inflammatory cytokines, misfolded proteins, or metabolic regulators, it may be possible to establish long-term immunological control over conditions like Alzheimer’s disease, atherosclerosis, and certain cancers [87]. For instance, antibodies originally developed to neutralize pathological amyloid-beta in Alzheimer’s treatment could inspire vaccine strategies that promote a sustained immune response to amyloid aggregation, potentially modifying the course of the disease. Converting antibody-based therapies into immunogenic formulations could serve as a cost-effective and scalable alternative to repeated monoclonal antibody administration in the management of chronic diseases.

### 7.2. Non-Biological

#### Behavioral

In addition to development challenges, the widespread adoption of NCD vaccines hinges on public willingness to accept them. Vaccine hesitancy, driven by fears about safety, mistrust in health authorities, and cultural or religious beliefs, can significantly limit uptake. This is especially relevant for NCD vaccines, which represent a novel prevention paradigm and may raise greater public uncertainty than traditional infectious disease vaccines [88].

Addressing this challenge requires a multifaceted approach grounded in transparent, evidence-based communication, delivered in ways that resonate across varying levels of scientific literacy. Trusted messengers, such as healthcare professionals, local leaders, and community organizations, should play a central role, as research consistently shows that health recommendations are more readily accepted when conveyed by familiar and credible sources [89]. To maximize impact, healthcare professionals should be trained not only in the relevant scientific knowledge but also in communication techniques that build trust and support informed decision-making [90]. Culturally tailored outreach is especially critical in conservative or under-resourced communities. For example, India has successfully engaged faith leaders as vaccination ambassadors, featuring them in personal story-driven videos to counter myths [91], while Pakistan has mobilized religious scholars, tribal elders, and female health workers to combat misinformation and promote immunization, particularly for polio and COVID-19 [92]. These models demonstrate how leveraging local credibility and cultural insight can be a powerful strategy for improving NCD vaccine acceptance and uptake.

Building on these community-based strategies, digital tools can further amplify accurate messaging and counter misinformation. Social media platforms and influencers can play a significant role in disseminating reliable vaccine information and debunking myths, thereby reducing public skepticism [93]. Transparency during the early stages of vaccine development is equally important; releasing safety data through credible institutions can strengthen trust and sustain acceptance over time. For NCD vaccination, actively applying these approaches can foster informed engagement and enhance public confidence.

Technological innovation can also play a role in reducing hesitancy. Non-invasive delivery methods—such as intranasal vaccines, microneedles, and patches—offer needle-free alternatives that are less intimidating and may be more acceptable in populations with injection-related fears. Studies show that intranasal vaccines, for instance, improve compliance, particularly among children and individuals whose cultural or religious practices discourage injections [94]. Accelerating the development and accessibility of these methods may significantly enhance uptake among hesitant populations. Therefore, as we invest in NCD vaccines, it is important to explore and implement alternative delivery methods that align with population preferences and help overcome barriers to acceptance.

To further improve equity and acceptance, targeted strategies must also address the needs of high-risk and marginalized groups. Vulnerable populations, including older adults, people with pre-existing conditions, and those in socioeconomically disadvantaged communities, often face barriers to vaccine access and health education [95]. Deploying mobile vaccination units and implementing localized education programs can help close these gaps. These efforts not only improve vaccine equity but also strengthen health systems more broadly for NCD prevention and care.

### 7.3. Funding

Despite progress in research and promotion strategies, the success of NCD vaccine development and adoption ultimately hinges on robust and strategic funding mechanisms. Currently, funding for NCDs is alarmingly low, comprising only 1–2% of global health financing [96]. This level of investment is insufficient, especially given the high costs associated with developing innovative vaccine technologies for NCD prevention. Sustainable Development Goal (SDG) 3.4, which aims to reduce premature NCD-related mortality by one-third by 2030, provides a global framework to guide investment priorities in NCD prevention efforts [97]. However, achieving this target will require a significant increase in funding for NCD research and vaccine innovation. Governments can optimize their health budgets by implementing measures such as taxing unhealthy commodities, eliminating subsidies to harmful industries, and improving the efficiency of service delivery. On a global scale, the international community should prioritize official development assistance (ODA) to support domestic resource mobilization in LMICs. ODA can play a catalytic role by filling initial funding gaps, supporting health system strengthening, and incentivizing local governments to invest in long-term NCD prevention strategies, including vaccine development and distribution. Institutions like the World Bank should provide technical guidance and funding recommendations to aid NCD investments. In particular, the World Bank can mobilize blended financing solutions, offer development policy loans tied to NCD-related targets, and support countries through results-based financing models [98].

A promising step in this direction is the Global NCD Compact 2020–2030, which outlines strategies for increased investment in NCD prevention, including the development of vaccines. It highlights funding tools such as the UN Multi-Partner Trust Fund and advocates for integrating NCD prevention into primary and universal healthcare systems [99]. Using this framework, governments can establish national policies to encourage NCD vaccine innovation and delivery.

In parallel, greater engagement from the private sector is critical to sustaining momentum in NCD vaccine development. Public–private partnerships, supported by tax incentives and subsidies, can leverage corporate investments and philanthropic contributions. Innovative financing models such as blended finance and social impact bonds may further accelerate research efforts. Establishing dedicated NCD vaccine funds could provide sustainable, long-term support. Gavi, the Vaccine Alliance, serves as a valuable model, having successfully attracted additional funding while promoting leadership and national ownership in vaccine delivery [100,101]. Entities focusing on NCD vaccine development can adopt similar innovative approaches.

### 7.4. Policy

#### 7.4.1. National Policies

After resolving funding constraints for NCD vaccine R & D, domestic governments should introduce policies aimed at improving vaccination rates. Mandating insurance coverage or providing subsidies can enhance accessibility, particularly for vulnerable populations, thereby increasing vaccine uptake. For instance, South Korea’s National Immunization Program provides free or subsidized influenza and pneumococcal vaccines to elderly and high-risk populations through insurance coverage and targeted subsidies, resulting in high vaccine uptake and improved public health outcomes [102]. Additional incentives, such as tax deductions, employer-sponsored vaccination initiatives, or conditional financial benefits, could further encourage participation [103]. According to a McKinsey report, employers such as Chobani, Dollar General, etc., have leveraged paid time off, financial incentives, cost reimbursements, and on-site vaccination to substantially increase COVID-19 vaccination uptake among employees [104]. Lessons learned from these global campaigns that successfully increase the vaccine uptake for infectious disease vaccines suggest potential pathways for the future promotion of NCD vaccines, emphasizing the importance of policy-level interventions.

#### 7.4.2. Global Collaboration

Given the varying epidemiology of NCDs across regions (Table 2), international collaboration is essential to fast-track vaccine development and equitable distribution. Establishing multi-center networks that allow countries to focus on their respective areas of expertise while sharing data and technologies could accelerate progress. Ensuring equitable access to vaccines is particularly important, as most research occurs in high-income countries while the burden of NCDs shifts increasingly toward LMICs [105]. To address this disparity, shared intellectual property mechanisms, supported by international agreements, could facilitate the licensing and distribution of vaccine technologies. Additionally, mutual recognition agreements (MRAs) between regulatory agencies would streamline approval processes, reducing costs and time delays. In this context, the WHO can play a pivotal role by setting global regulatory standards, promoting regulatory convergence through initiatives such as the Prequalification Programme and the Global Benchmarking Tool, and providing technical assistance and capacity-building to strengthen regulatory systems in China and other LMICs [106].

### 7.5. Structural

Sustainable NCD vaccine promotion will require interventions at the structural level. Integrating vaccine delivery into established healthcare systems, such as maternal and child health programs or chronic disease management services, can simplify logistics while reinforcing the role of vaccination as part of routine preventive care [107].

Digital health infrastructure also has a significant role in enhancing delivery and tracking. Governments should adopt nationwide electronic health records (EHRs) and immunization registries to systematically track vaccinations, deliver automated reminders, and identify gaps in coverage. Digital systems, combined with big data analytics, can improve supply chain management, predict vaccine demand, and enable targeted outreach [108]. To maximize the impact of these digital innovations, WHO can take an active role in fostering data sharing frameworks and supporting pooled procurement mechanisms for NCD vaccines, leveraging existing models like COVAX [109]. Such approaches can significantly reduce missed vaccinations and increase compliance, particularly in large-scale immunization efforts.

Despite these advancements, gaps remain in global NCD vaccine development, especially in countries like China. While developed nations with advanced research ecosystems make rapid progress in NCD vaccine innovation, countries like China face challenges such as regulatory delays and limited resources. To overcome these barriers, China should increase investment in research infrastructure and foster collaboration between academia and industry. Streamlining regulatory approval processes would also facilitate faster translation of research into accessible vaccines. China and other adjacent developing countries should learn from the African Medicines Regulatory Harmonization (AMRH) program and the ASEAN Joint Assessment, as they enhance local regulatory efficiency by enabling joint dossier reviews, streamlining approval pathways, and promoting mutual recognition among neighboring countries [110]. Together, these national and regional mechanisms can reduce regulatory fragmentation and accelerate the availability of NCD vaccines in underserved settings.

Cuba serves as another excellent example for developing countries like China. Despite economic challenges, Cuba achieved remarkable progress with CIMAvax-EGF, a therapeutic vaccine for lung cancer. Its success stemmed from early prioritization of biotechnology, integration of research and manufacturing, and alignment of scientific goals with public health needs [19]. Developing nations can follow Cuba’s model by investing in biotechnology, integrating research with production capabilities, and focusing on vaccines addressing diseases prevalent domestically. This pragmatic approach could ensure equitable progress in NCD vaccine development globally.

## 8. Conclusions

The global spread of NCDs and the heavy burden it imposes on China’s public health system urgently call for innovative solutions that go beyond traditional treatment models. This review analysis indicates that therapeutic vaccines have demonstrated transformative potential in intervening with chronic diseases, especially in the field of cancer. However, there are multiple significant obstacles on its road from concept to wide application. The primary challenge lies in the mismatch between the focus of vaccine R & D and the real burden of diseases: despite the progress made in cancer vaccines, there is a lack of targeted investment in vaccine R & D for key NCDs that cause huge health losses in China, such as stroke, chronic obstructive pulmonary disease (COPD), and chronic kidney disease. Secondly, the problem of unbalanced regional development is prominent: as one of the countries with the heaviest burden of NCDs globally, China lags significantly in the independent R & D of key therapeutic vaccines, and the existing gap in medical resources between urban and rural areas further exacerbates the inequality in the accessibility of innovative technologies. Finally, the transformation bottleneck cannot be ignored: the complex multifactor etiology of NCDs, the strict requirements for the long-term safety and efficacy verification of vaccines, and the potential hesitation of the public towards such new preventive/therapeutic methods jointly constitute the substantial obstacles for vaccines to move from the laboratory to clinical practice and a wide population.

To overcome these obstacles, it is necessary to construct a multi-dimensional and cross-disciplinary collaborative action framework. At the level of scientific exploration, the utilization of artificial intelligence to drive the integration of multi-omics data should be accelerated to deeply explore disease-specific targets, such as exploring the regulatory role of immune checkpoints in cardiovascular diseases and neurodegenerative diseases. At the same time, actively promoting the transformation of existing successful antibody therapies into active immune strategies will also play a crucial role. At the implementation and application level, it is necessary to establish a precise vaccination system based on risk stratification and fully integrate electronic health records to achieve dynamic identification and management of high-risk groups. In addition, efforts should be made to develop non-invasive delivery technologies (such as nasal sprays and microneedle patches) to enhance patient compliance and acceptance. At the level of policy support, China should give priority to laying out the R & D of vaccines targeting the domestic high-incidence disease spectrum, and ensure the continuous investment in R & D through innovative financing mechanisms. At the global level, it is urgently necessary to establish a fair intellectual property rights sharing mechanism and rely on the coordination of the World Health Organization (WHO) to promote mutual recognition of regulations and accelerate the accessibility of vaccines in resource-limited areas.

The ultimate significance of NCD vaccines goes far beyond a single technological breakthrough; it lies more in their potential to reshape the overall prevention and control paradigm of chronic diseases, from passively responding to advanced symptoms to proactively building early immune defenses. For China, if it can make concerted efforts in the three key stages of target discovery, cross-regional clinical trial collaboration, and digital health infrastructure construction, it will not only effectively promote the realization of the “Healthy China 2030” strategic goal but also may become the core force leading the global process of fair NCD vaccine. Provide crucial technological support for the commitment to the United Nations Sustainable Development Goals (SDGs) to reduce the premature mortality rate of NCDs. The success of this transformation requires close collaboration among scientific research, industry, the government and society to jointly open a new chapter in the prevention and control of chronic diseases.

## Figures and Tables

**Figure 1 vaccines-13-00881-f001:**
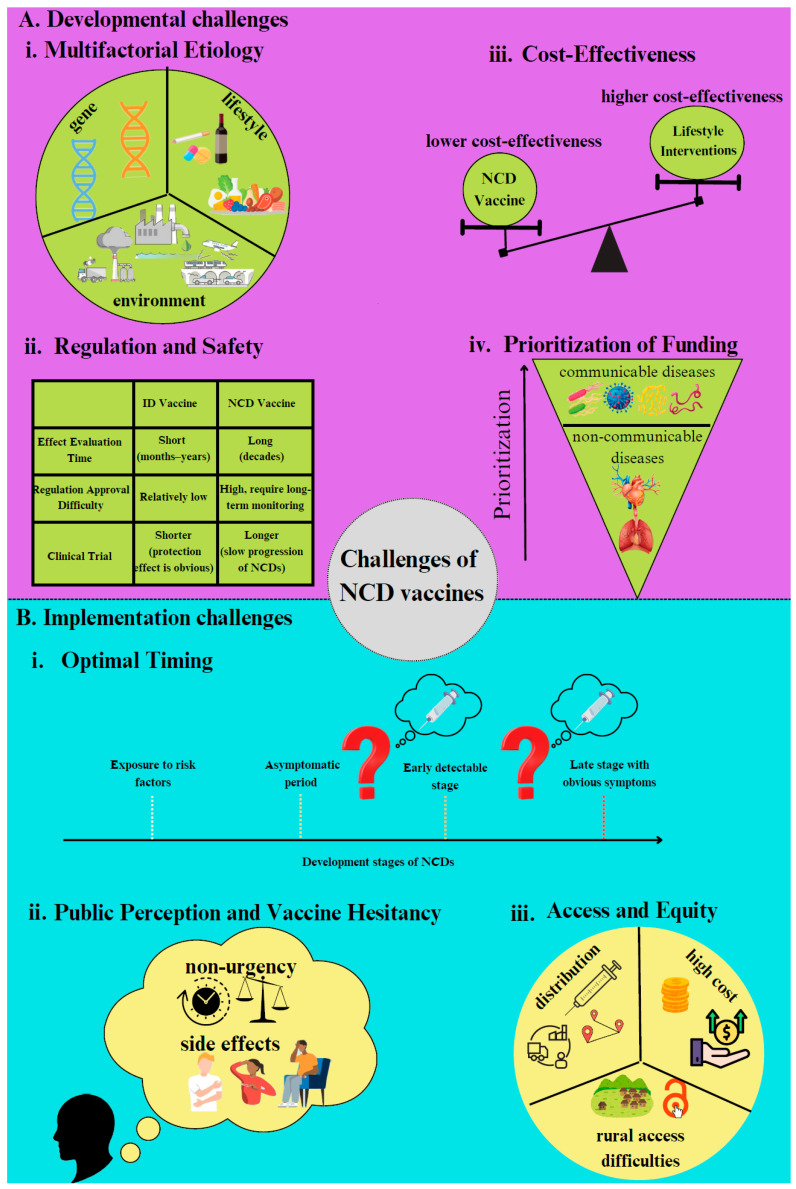
Key developmental and implementation challenges in advancing vaccines for non-communicable diseases (NCDs): A visual representation of barriers hindering both the development and implementation of vaccines for NCDs, highlighting critical scientific, regulatory, logistical, and societal obstacles. (**A**). Developmental Challenges: This diagram outlines the primary obstacles hindering the development of vaccines for non-communicable diseases (NCDs), identifying four key challenges: multifactorial etiology, regulatory and safety requirements, cost-effectiveness, and funding prioritization. The first section highlights the complexity of NCDs, which arise from a combination of genetic, behavioral, and environmental factors, making it difficult to pinpoint a single immunological target. This multifactorial nature contrasts with infectious diseases, where vaccines often target specific pathogens, thus complicating traditional vaccine development models. The second section addresses the heightened regulatory and safety requirements for NCD vaccines, necessitating extensive long-term efficacy data and safety monitoring. Unlike infectious disease vaccines, which often yield outcomes quickly, NCD vaccines require years to demonstrate their therapeutic impact, and clinical trials can be easily terminated because of a lack of efficiency. The third section focuses on the challenge of cost-effectiveness, as preventive lifestyle changes often offer better outcomes at lower costs, making large-scale investment in NCD vaccines difficult to justify. The fourth section emphasizes how limited funding and a lack of prioritization impede progress in NCD vaccine research, as global health funding tends to focus on communicable diseases with immediate public health threats, leaving NCDs underfunded despite their significant global burden. Together, these challenges highlight the need for innovative scientific approaches, equitable resource allocation, and sustained commitment to drive progress in NCD vaccine development. (**B**). Implementation Challenges: This diagram illustrates the primary challenges associated with implementing NCD vaccines, identifying three major obstacles: optimal timing, public perception and vaccine hesitancy, and access and equity. The first challenge involves determining the appropriate timing for vaccination. NCDs often develop gradually and with subtle symptoms, making early detection difficult. This uncertainty complicates identifying the most effective stage for intervention. Additionally, the slow progression of NCDs makes it challenging to pinpoint the ideal window for vaccination to achieve maximum impact. The second challenge focuses on public perception and vaccine hesitancy. Unlike infectious disease vaccines, NCD vaccines may not be viewed as urgent or immediately necessary due to the long-term nature of NCD development. This lack of perceived urgency, coupled with concerns about potential side effects, particularly in regions with a history of vaccine safety issues, contributes to hesitancy among populations. The final challenge relates to access and equity, encompassing three critical barriers: affordability, distribution issues, and rural accessibility. The high cost of NCD vaccines often makes them inaccessible for lower-income populations. Distribution challenges, including ineffective supply chains and logistical complexities, further limit availability. In rural and remote areas, weak healthcare infrastructure exacerbates these difficulties, restricting widespread vaccine coverage. These interconnected challenges highlight the complexities involved in successfully integrating NCD vaccines into public health initiatives, underscoring the need for innovative solutions and equitable healthcare strategies.

**Figure 2 vaccines-13-00881-f002:**
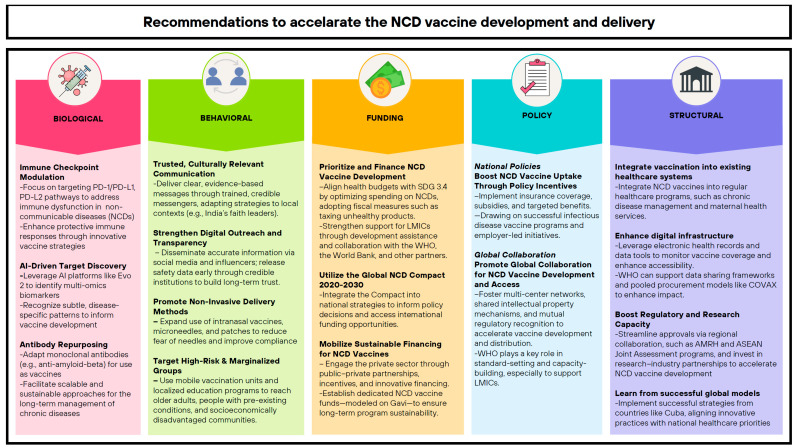
A schematic diagram of comprehensive recommendations to accelerate NCD vaccine development and delivery across four domains. This schematic highlights the following: (1) Biological innovations: Targeting immune checkpoints (e.g., PD-1/PD-L1, PD-L2) to restore immune function in NCDs, using AI-driven multi-omics platforms (e.g., Evo 2) for biomarker discovery, and repurposing monoclonal antibodies (e.g., anti-amyloid-beta) as scalable vaccine candidates. (2) Behavioral strategies: Building public trust through culturally adapted evidence-based messaging delivered by credible messengers, strengthening digital outreach and transparency, promoting non-invasive delivery methods (e.g., microneedles, patches), and implementing targeted outreach to marginalized and high-risk populations via mobile units and localized education. (3) Funding measures: Aligning health budgets with SDG 3.4 goals, implementing fiscal policies such as taxing unhealthy products, mobilizing sustainable financing through public–private partnerships, leveraging the Global NCD Compact 2020–2030, and establishing dedicated vaccine funds modeled after Gavi. (4) Policy and structural approaches: Enhancing national uptake through insurance coverage, subsidies, and targeted benefits; fostering global collaboration via multi-center R & D networks, shared intellectual property, and regulatory convergence with WHO support; integrating vaccination into existing healthcare services; strengthening digital health infrastructure for tracking and supply chain optimization; boosting regulatory and research capacity through regional collaboration (e.g., AMRH, ASEAN Joint Assessment); and adopting successful global models such as Cuba’s integrated biotech and public health strategy. Together, these measures form a multi-dimensional framework to close translational gaps and ensure equitable global deployment of NCD vaccines.

**Table 1 vaccines-13-00881-t001:** Top 10 non-communicable diseases * by DALYs and death percentage in China and globally (2021) [2].

	By DALYs				By Death			
Rank	China	DALYs %	Global	DALYs %	China	Death %	Global	Death %
1	Stroke	13.23%	Ischemic heart disease	6.55%	Stroke	22.16%	Ischemic heart disease	13.25%
2	Ischemic heart disease	8.87%	Stroke	5.57%	Ischemic heart disease	16.73%	Stroke	10.68%
3	COPD	5.88%	COPD	2.77%	COPD	10.99%	COPD	5.48%
4	Tracheal, bronchus, and lung cancer	4.7%	Diabetes mellitus	2.73%	Tracheal, bronchus, and lung cancer	6.95%	Tracheal, bronchus, and lung cancer	2.97%
5	Age-related and other hearing loss	3.07%	Low back pain	2.42%	Alzheimer’s disease and other dementias	4.22%	Alzheimer’s disease and other dementias	2.88%
6	Diabetes mellitus	2.9%	Depression disorders	1.95%	Stomach cancer	3.8%	Diabetes mellitus	2.44%
7	Low back pain	2.8%	Congenital birth defects	1.82%	Hypertensive heart disease	2.81%	Chronic kidney disease	2.25%
8	Stomach cancer	2.64%	Headache disorders	1.65%	Esophageal cancer	2.53%	Cirrhosis and other chronic liver diseases	2.1%
9	Alzheimer’s disease and other dementias	2.5%	Tracheal, bronchus, and lung cancer	1.62%	Colon and rectum cancer	2.35%	Hypertensive heart disease	1.96%
10	Depression disorders	1.95%	Cirrhosis and other chronic liver diseases	1.61%	Chronic kidney disease	1.75%	Colon and rectum cancer	1.54%

* Definitions of non-communicable diseases are based on and supported by the Global Burden of Disease (GBD) website.

**Table 2 vaccines-13-00881-t002:** Summary of existing therapeutic vaccines for major non-communicable diseases (globally and in China): examples and stages *.

NCD	Example	Stages in Other Countries	Stages in China
Lung cancer	CimaVax-EGF (Cuba)	Approved for Market ^1^	Clinical trial ^2^
Liver cancer	GPC3 (Japan)	Phase I/Phase II ^3^	Phase I/II
Gastric cancer	HER-Vaxx (Austria)	Phase Ib/II ^4^ (Terminated)	Phase Ib/II ^4^
Pancreatic cancer	CellgramDC-WT1 (Japan)	Phase I/II ^5^	NA
Colorectal cancer	OncoVAX (United States)	Phase III ^6^	NA
Kidney cancer	NeoVax (United States)	Phase I ^7^	NA
Prostate cancer	Sipuleucel-T (United States)	Approved for Market ^8^	NA
Hypertension	CYT006-AngQb (Switzerland)	Phase IIa ^9^ (Discontinued)	NA
Dyslipidemia	Inclisiran (Switzerland)	Approved for Market ^10^	Approved for market ^10^
Diabetes mellitus	CYT013-IL1bQb (Switzerland)	Phase I/IIa ^11^	NA
Alzheimer’s disease	CAD106	Phase II/III ^12^	NA

* Note: The information in this table is derived from several articles [18,19,20,21,22,23,24,25,26,27,28] and websites [29,30,31,32,33,34,35,36,37,38], collected up to 8 July. ^1^ CimaVax-EGF is now an approved treatment for lung cancer in Cuba, Argentina, Bosnia and Herzegovina, Colombia, Kazakhstan, Paraguay, and Peru. Meanwhile, clinical trials have been performed in Japan, the United States, and other countries. ^2^ No specific stage and progress in the clinical trial were reported; the clinical result is unknown. ^3^ Japan has finished the phase II study of the GPC3-derived peptide vaccine; the United States is still recruiting participants for their phase I trial. ^4^ The HERIZON Trial (NCT02795988) is a phase II trial located in Georgia, India, Moldova, Serbia, Thailand, Ukraine, and Taiwan, China. However, the nextHERIZON Trial (NCT05311176), located in Australia and Taiwan, China, was terminated because of slow enrollment and organizational priorities. ^5^ Japan remains the only country with registered clinical trials involving CellgramDC-WT1, with both phase I/IIa and phase II trials. ^6^ The United States remains the only country with registered clinical trials involving OncoVax. ^7^ The United States is the only country with registered clinical trials involving NeoVax. ^8^ Sipuleucel-T received FDA approval in 2010. Although it was granted marketing authorization by the EMA in 2013 for men with few or no symptoms, the manufacturer later requested its withdrawal in 2015. A phase III clinical trial was performed in the United States and Canada. ^9^ Available records indicate that CYT006-AngQb has only undergone a phase IIa clinical trial in Switzerland. Phase III trials were not conducted because of its lower antihypertensive effects compared to existing RAS inhibitors [39]. ^10^ Inclisiran has been approved in the European Union, the United Kingdom, the United States, and China. Its phase III clinical trials were conducted in 13 countries, including Canada, the Czech Republic, Denmark, etc. The phase IV trial in China, which aims to evaluate the real-world effectiveness of Inclisiran relative to standard care in Chinese patients, is currently undergoing recruitment [40]. ^11^ Phase I/IIa trials were conducted in Germany and Switzerland. ^12^ Phase II/III trials were conducted in the United States, Australia, Belgium, Canada, Finland, Germany, the Netherlands, Spain, Switzerland, and the United Kingdom.

## Data Availability

No new data were created in this review article.

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
