# Peer review of "Therapeutic Vaccines for Non-Communicable Diseases: Global Progress and China’s Deployment Pathways"

_vaccines, 2025, doi:10.3390/vaccines13080881_

Round 1
Reviewer 1 Report
Comments and Suggestions for Authors
- The data provided by the Global Burden of Disease (GBD) 2021 should be updated.
- A bibliographic citation must be provided. It should be related to the information summarized in Table 1.
- A bibliographic citation must be provided related to the following paragraph: “Traditionally used to combat communicable diseases, vaccination is now being considered as a potential strategy for addressing NCDs”.
- A bibliographic citation must be provided related to the following paragraph: “although other conditions like 103 hypertension and Alzheimer’s disease are also being investigated”.
- What is the mechanism of action of vaccines intended for COPD or Chronic kidney disease?
- The authors should reference the mechanism of action of vaccines in the clinical phase to combat the non-cancerous diseases they refer to in paragraphs 160-172.
- The authors should comment on China's lack of interest in NCD vaccine development.
- The authors should explain that NCD vaccines require years to demonstrate their preventive impact
Author Response
Reviewer #1 (Remarks to the Author):
The data provided by the Global Burden of Disease (GBD) 2021 should be updated.
Response: Thank you for your feedback. After repeated confirmation through the Institute for Health Metrics and Evaluation (IHME) website, despite our best efforts to use the most up-to-date data, the official GBD database is currently only updated to 2021.
A bibliographic citation must be provided. It should be related to the information summarized in Table 1.
Response: Thank you for your suggestion. Since all the content included in Table 1 was derived from one single source - the GBD 2021 database (IHME), We have already added a citation to the title of Table 1 shown as below:
Table 1. Top 10 Non-Communicable Diseases* by DALYs and Death Percentage in China and Globally (2021) (2).
A bibliographic citation must be provided related to the following paragraph: “Traditionally used to combat communicable diseases, vaccination is now being considered as a potential strategy for addressing NCDs”.
Response: Thank you for the valuable suggestion, we have now added an appropriate bibliographic citation to support the statement, see Page 3, Line 69 in the revised manuscript.
A bibliographic citation must be provided related to the following paragraph: “although other conditions like hypertension and Alzheimer’s disease are also being investigated”.
Response: Thank you for your feedback, we have now added two addtional bibliographic citations to highlight clinical investigations conducted on vaccines for hypertension and Alzheimer’s disease on Page 4, Line 133
What is the mechanism of action of vaccines intended for COPD or Chronic kidney disease?
Response: We appreciate your concern regarding the mechanisms of action of vaccines intended for COPD and chronic kidney disease. According to the GBD data, both COPD and CKD rank among the leading contributors to DALYs and deaths globally and in China, underscoring their public health significance. However, as no vaccines targeting these conditions have yet been formally developed, we could only identify potential vaccine development approaches, which remain at a very early research stage and require substantial additional experimental data to establish their credibility. For this reason, we intentionally kept the discussion in the main text brief, while citing references that provide detailed descriptions of these potential mechanisms on Page 6, Lines 210-213
The authors should reference the mechanism of action of vaccines in the clinical phase to combat the non-cancerous diseases they refer to in paragraphs 160-172.
Response: We appreciate your suggestion. In the revised manuscript, we have expanded the discussion in the relevant section to include additional vaccines listed in Table 2, as well as therapeutic vaccines not previously mentioned. For each, we have briefly described their mechanisms of action and provided the corresponding citations (See Page 6, Lines 217–234).
The authors should comment on China's lack of interest in NCD vaccine development.
Response: Thank you for such an inspiring suggestion. We agree that it is important to assess whether China shows a lack of interest in NCD vaccine development and to explore the underlying reasons. Accordingly, in the revised manuscript, we have added an analysis at the end of the “Developmental Challenges” section, supported by specific data, which concludes that China currently demonstrates limited interest in NCD vaccines. This addition also serves as a basis for the recommendations presented in the subsequent section (see Page 10, Lines 364–377).
The authors should explain that NCD vaccines require years to demonstrate their preventive impact
Response: Thank you for your suggestion. We have added supporting evidence as well as specific examples to explain the prolonged timeline for NCD vaccines’ development in the “Developmental Challenge” section, in the paragraph addressing regulatory and safety considerations (Page 9, Lines 326-336). What is more, we need to clarify in this manuscript, instead of focusing NCD vaccines' preventive impact, we are more interested in their therapeutic impact. Therefore, in the main text, we have also changed this sentence to “NCD vaccines require years to demonstrate their preventive impact” (Line 216)
Reviewer 2 Report
Comments and Suggestions for Authors
Excellent paper - very well written - comprehensive and very very relevant.
Fascinating explanations and diagrammatic representations of situation in research, implementation and recommendations going forward.
TOtally agree with the issues of some NCDs like cancer getting undue priority compared to more prevalent diseases.
Congratulations to the team.
Well done
Author Response
Reviewer #2 (Remarks to the Author):
Excellent paper - very well written - comprehensive and very very relevant. Fascinating explanations and diagrammatic representations of situation in research, implementation and recommendations going forward. Totally agree with the issues of some NCDs like cancer getting undue priority compared to more prevalent diseases. Congratulations to the team. Well done.
Response:Thank you very much for your kind and encouraging feedback. We truly appreciate your positive comments on the clarity, relevance, and comprehensiveness of our work. Your support is deeply appreciated by our entire team.
Reviewer 3 Report
Comments and Suggestions for Authors
The manuscript titled “Global Progress and Challenges in Non-communicable Diseases Vaccines: A Focused Analysis on China's Disease Burden and Deployment Strategies” discusses an important and timely topic—therapeutic and prophylactic vaccines for non-communicable diseases (NCDs)—with a particular focus on China. The authors provide a thorough overview of global progress and challenges, emphasizing the disconnect between disease burden and vaccine development priorities, as well as the uneven pace of innovation and implementation across different countries.
The paper is ambitious in scope, well-structured, and generally well-written. However, several issues need to be addressed before the manuscript can be considered for publication.
I recommend major revisions.
- The title is informative, but too long. Consider shortening it to improve clarity and impact.
- Although the manuscript reads like a narrative review, some methodological details (such as the database searched, inclusion criteria, and time window) should be reported to improve transparency and reproducibility;
- While the focus on China is valuable, the global overview is more superficial. A better balance is needed between international trends and the Chinese context. For example, the comparison with other LMICs could be expanded;
- Tables 1 and 2 are helpful but would benefit from clearer formatting and more explicit explanation in the main text.
- The manuscript refers to “NCD vaccines” in both prophylactic and therapeutic contexts. It would be helpful to define these terms more clearly and specify their usage throughout the manuscript.
- Some sections, especially sections 4 and 5, are repetitive. The challenges of funding, regulatory hurdles, and vaccine hesitancy appear multiple times. Merging these sections would enhance clarity and readability.
- Several citations are appropriate, but some claims—particularly about funding distribution and public perception—would benefit from stronger or more recent supporting data.
- Figures 1 and 2 could be helpful, but lack sufficient resolution and clarity. Please improve the graphics quality and make sure the legends are self-explanatory—Table 2 mixes approved and clinical-stage vaccines. Consider separating them or adding a column to clarify the development stage and regulatory status.
- The recommendation section is one of the strongest parts of the paper, offering a clear framework. However, the recommendations would benefit from greater specificity (e.g., naming successful models or best practices beyond Cuba);
- The potential role of WHO, Gavi, and regional regulatory harmonization initiatives could be further elaborated.
- The English language is generally acceptable, but some parts are wordy or have awkward phrasing. Professional editing would be recommended.
Author Response
Reviewer #3 (Comments to the Author):
The manuscript titled “Global Progress and Challenges in Non-communicable Diseases Vaccines: A Focused Analysis on China's Disease Burden and Deployment Strategies” discusses an important and timely topic—therapeutic and prophylactic vaccines for non-communicable diseases (NCDs)—with a particular focus on China. The authors provide a thorough overview of global progress and challenges, emphasizing the disconnect between disease burden and vaccine development priorities, as well as the uneven pace of innovation and implementation across different countries.
The paper is ambitious in scope, well-structured, and generally well-written. However, several issues need to be addressed before the manuscript can be considered for publication.
Response: We thank the reviewer for the positive comments.
- The title is informative, but too long. Consider shortening it to improve clarity and impact.
Response: We changed it into “Therapeutic Vaccines for Non-Communicable Diseases: Global Progress and China's Deployment Pathways”, expressing our main focus on therapeutic NCDs vaccines while looking at China specifically.
- Although the manuscript reads like a narrative review, some methodological details (such as the database searched, inclusion criteria, and time window) should be reported to improve transparency and reproducibility;
Response: Thank you for this important comment! To address this, we have now added a dedicated Methodology section in the revised manuscript to improve transparency and reproducibility. This section clearly specifies the literature review design, databases searched, keyword combinations, inclusion criteria, and the time window for data collection. It also details the types of sources consulted, including regulatory databases, national policy documents, and market intelligence reports, as well as the internal review process for ensuring source credibility (see Page 3, Lines 91–120).
- While the focus on China is valuable, the global overview is more superficial. A better balance is needed between international trends and the Chinese context. For example, the comparison with other LMICs could be expanded;
Response: Thank you for your suggestion. We have expanded the global perspective by adding examples of LMICs in multiple sections of the manuscript:(1)In Section 6 Implementation Challenges, added Nigeria as an example of a country lacking sufficient screening health facilities, illustrating that LMICs face challenges similar to those in China (Page 11, Lines 404–407). (2) In Recommendations section, 6.2 Non-biological, Behavioral, added India and Pakistan as examples where faith leaders and religious scholars have successfully served as vaccination ambassadors to reduce vaccine hesitancy, suggesting that similar approaches could be applied to NCD vaccination campaigns (Page 14, Lines 526–532). (3)In Recommendations section, 6.5 Structural, provided further details on how Regional Harmonization Initiatives programs such as the African Medicines Regulatory Harmonization (AMRH) initiative and the ASEAN Joint Assessment improve regulatory efficiency, particularly in LMICs (Page 16, Lines 645–651).
- Tables 1 and 2 are helpful but would benefit from clearer formatting and more explicit explanation in the main text.
Response: We appreciate this thoughtful insight. We have revised both Table 1 and Table 2 for improved clarity and informativeness as shown below. Specifically, in Table 1, we separated DALYs and Deaths into an additional row, making the information more visually accessible. In Table 2, we removed the coarse classification of “medical use” and “market use” and replaced it with precise clinical phases. This adjustment provides more valuable and comparable information, facilitating a clearer comparison between China and other countries.
Moreover, we have enriched the main text description for both tables. For Table 1, we added further explanations in the relevant section, with particular emphasis on COPD. We noted that stroke, heart disease, and COPD rank among the top three in both global and Chinese DALYs and deaths. While the main text previously discussed stroke and heart disease, we have now incorporated a more detailed interpretation of COPD data, along with commentary on certain other NCDs that were not previously addressed. (See Page 2, Lines 47-52)
For Table 2, we expanded the discussion in the existing NCD vaccines section to provide a more granular comparison between China and other countries(See Page 6-7, Lines 239-246), and in the developmental challenges section to include more specific vaccines listed in the table, noting cases where development was terminated or discontinued.(See Page 9, Lines 314-322) Additionally, we added detailed explanations and notes directly beneath Table 2 to further clarify the data presented. (See Page 5, Lines 149-175)
Table 1
|
By DALYs |
By Death |
||||||
Rank |
China |
DALYs % |
Global |
DALYs % |
China |
Death % |
Global |
Death % |
1 |
Stroke |
13.23% |
Ischemic heart disease |
6.55% |
Stroke |
22.16% |
Ischemic heart disease |
13.25% |
2 |
Ischemic heart disease |
8.87% |
Stroke |
5.57% |
Ischemic heart disease |
16.73% |
Stroke |
10.68% |
3 |
COPD |
5.88% |
COPD |
2.77% |
COPD |
10.99% |
COPD |
5.48% |
4 |
Tracheal, bronchus, and lung cancer |
4.7% |
Diabetes mellitus |
2.73% |
Tracheal, bronchus, and lung cancer |
6.95% |
Tracheal, bronchus, and lung cancer |
2.97% |
5 |
Age-related and other hearing loss |
3.07% |
Low back pain |
2.42% |
Alzheimer’s disease and other dementias |
4.22% |
Alzheimer’s disease and other dementias |
2.88% |
6 |
Diabetes mellitus |
2.9% |
Depression disorders |
1.95% |
Stomach cancer |
3.8% |
Diabetes mellitus |
2.44% |
7 |
Low back pain |
2.8% |
Congenital birth defects |
1.82% |
Hypertensive heart disease |
2.81% |
Chronic kidney disease |
2.25% |
8 |
Stomach cancer |
2.64% |
Headache disorders |
1.65% |
Esophageal cancer |
2.53% |
Cirrhosis and other chronic liver diseases |
2.1% |
9 |
Alzheimer’s disease and other dementias |
2.5% |
Tracheal, bronchus, and lung cancer |
1.62% |
Colon and rectum cancer |
2.35% |
Hypertensive heart disease |
1.96% |
10 |
Depression disorders |
1.95% |
Cirrhosis and other chronic liver diseases |
1.61% |
Chronic kidney disease |
1.75% |
Colon and rectum cancer |
1.54% |
Table 2
NCD |
Example |
Stages in Other Countries |
Stages in China |
Lung cancer |
CimaVax-EGF (Cuba) |
Approved for market1 |
Clinical trial2 |
Liver cancer |
GPC3 (Japan) |
Phase I/Phase II3 |
Phase I/II |
Gastric cancer |
HER-Vaxx (Austria) |
Phase Ib/II4 (Terminated) |
Phase Ib/II4 |
Pancreatic cancer |
CellgramDC-WT1 (Japan) |
Phase I/II5 |
NA |
Colorectal cancer |
OncoVAX (United States) |
Phase III6 |
NA |
Kidney cancer |
NeoVax (United States) |
Phase I7 |
NA |
Prostrate cancer |
Sipuleucel-T (United States) |
Approved for Market8 |
NA |
Hypertension |
CYT006-AngQb (Switzerland) |
Phase IIa9 (Discontinued) |
NA |
Dyslipidemia |
Inclisiran (Switzerland) |
Approved for Market10 |
Approved for market10 |
Diabetes mellitus |
CYT013-IL1bQb (Switzerland) |
Phase I/IIa11 |
NA |
Alzheimer’s disease |
CAD106 (Switzerland) |
Phase II/III12 (Terminated) |
NA |
- The manuscript refers to “NCD vaccines” in both prophylactic and therapeutic contexts. It would be helpful to define these terms more clearly and specify their usage throughout the manuscript.
Response: We appreciate the reviewer’s comment. Given that our manuscript primarily focuses on therapeutic vaccines for NCDs rather than prophylactic vaccines, we believe it is neither necessary nor feasible to refer to prophylactic vaccines throughout the text, especially since there are currently almost no existing prophylactic NCD vaccines to discuss. To address this concern and avoid ambiguity, we have revised the manuscript title to explicitly emphasize our focus on therapeutic NCD vaccines. We have also ensured that the term “therapeutic NCD vaccines” is used consistently throughout the manuscript where appropriate, including in the title.
- Some sections, especially sections 4 and 5, are repetitive. The challenges of funding, regulatory hurdles, and vaccine hesitancy appear multiple times. Merging these sections would enhance clarity and readability.
Response: Thank you for this suggestion. We have revised the structure to reduce repetition and improve clarity. Several paragraphs in Section 5 have been merged, and the order has been adjusted for greater logical flow. The revised Section 5 Development Challenges now focuses on the multifactorial nature of NCD vaccine development, regulatory and safety concerns, low cost-effectiveness, and lack of funding.(Page 9-10, Lines 303-382) Section 6 Implementation Challenges now concentrates on the challenges of optimal timing, vaccine hesitancy, and uneven access.(Page 10-11, Lines 383-439)
- Several citations are appropriate, but some claims—particularly about funding distribution and public perception—would benefit from stronger or more recent supporting data.
Response: Thank you for this suggestion. In response, we have strengthened both the recency and robustness of the evidence base for the relevant claims. For funding distribution parts, we added the latest NCD Alliance 2025 data (Page 10, Lines 355–357) and supplemented it with additional sources on China’s funding distribution, underscoring the relatively low investment in NCD vaccines, particularly cancer vaccines, compared to global trends (Page 10, Lines 364–377). These updates provide a more up-to-date and globally contextualized picture. Moreover, for public perception parts, we updated the supporting evidence by including recent data from India, illustrating the post–COVID-19 vaccine hesitancy (Page 11, Lines 419–422). This strengthens the claim with a current, concrete example from an LMIC context.
- Figures 1 and 2 could be helpful, but lack sufficient resolution and clarity. Please improve the graphics quality and make sure the legends are self-explanatory—Table 2 mixes approved and clinical-stage vaccines. Consider separating them or adding a column to clarify the development stage and regulatory status.
Response: Thank you for your insightful comment. For Figure 1, we have adjusted the sequence of “developmental challenges” to align with the revised main text, and for “implementation challenges,” we have modified the representation of “optimal timing” to make it clearer and avoid potential misinterpretation. We have also increased the font size to enhance readability. (See attachment PDF file)
For Figure 2, we made minor adjustments to reflect the updated manuscript content and changed the subheading font color from white to black to improve clarity and visual appeal. (See attachment PDF file)
Regarding Table 2, as addressed in our earlier response, we have modified the columns to clearly indicate the development stage and regulatory status, thereby resolving the mixing of approved and clinical-stage vaccines.
- The recommendation section is one of the strongest parts of the paper, offering a clear framework. However, the recommendations would benefit from greater specificity (e.g., naming successful models or best practices beyond Cuba);
Response:Thank you for this valuable suggestion. We have incorporated additional specific examples to strengthen the recommendation section:(1)Added India and Pakistan as examples of successful strategies for addressing vaccine hesitancy (Page 14, Lines 526-531). (2) Added Korea as an example of improving vaccination rates, which could inform NCD vaccination promotion strategies (Page 15, Lines 597-600). (3)Expanded the section on WHO and Gavi initiatives to further enhance the specificity of our recommendations. (specific details see response to the next comment)
- The potential role of WHO, Gavi, and regional regulatory harmonization initiatives could be further elaborated.
Response: Thank you for this suggestion. We have expanded our discussion of the potential roles of key global and regional actors as follows:(1)World Bank: Further elaborated on how the World Bank can play a critical role in facilitating NCD vaccine funding (Page 15, Lines 572-576). (2)WHO: Expanded on WHO’s role in facilitating data sharing for NCD vaccines (page 16, line 634-636) and supporting the regulatory process (Page 16, Lines 619-623). (3)Regional Harmonization Initiatives: Provided additional details on how programs such as the African Medicines Regulatory Harmonization (AMRH) initiative and the ASEAN Joint Assessment improve regulatory efficiency, particularly in LMICs (Page 16, Lines 645-651).
- The English language is generally acceptable, but some parts are wordy or have awkward phrasing. Professional editing would be recommended.
Response: Thank you for your feedback. We have read through the entire text and corrected some minor wording. To clarify, all of our content is self-generated, except for the use of grammarly for grammar correction and sentence smoothness. Please let me know if any specific areas require further improvement.

Round 2
Reviewer 1 Report
Comments and Suggestions for Authors
The authors addressed the reviewer's concerns satisfactorily.
Reviewer 3 Report
Comments and Suggestions for Authors
None